# Structure of an MHC I–tapasin–ERp57 editing complex defines chaperone promiscuity

Ines Katharina Müller [1], Christian Winter [1], Christoph Thomas [1], Robbert M. Spaapen[2,3], Simon Trowitzsch [1] ✉ & Robert Tampé [1] ✉

Adaptive immunity depends on cell surface presentation of antigenic peptides by major histocompatibility complex class I (MHC I) molecules and on stringent ER quality control in the secretory pathway. The chaperone tapasin in conjunction with the oxidoreductase ERp57 is crucial for MHC I assembly and for shaping the epitope repertoire for high immunogenicity. However, how the tapasin–ERp57 complex engages MHC I clients has not yet been determined at atomic detail. Here, we present the 2.7-Å crystal structure of a tapasin–ERp57 heterodimer in complex with peptide-receptive MHC I. Our study unveils molecular details of client recognition by the multichaperone complex and highlights elements indispensable for peptide proofreading. The structure of this transient ER quality control complex provides the mechanistic basis for the selector function of tapasin and showcases how the numerous MHC I allomorphs are chaperoned during peptide loading and editing.

The presentation of antigenic peptides by major histocompatibility complex class I (MHC I) molecules on the cell surface ensures the detection and elimination of infected or cancerous cells by the adaptive immune system. Any misinterpretation can lead to inflammatory and autoimmune diseases. Cytotoxic T lymphocytes, which are licensed to kill, search for non-self peptides presented by MHC I[1,2]. MHC I complexes are composed of a heavy chain (hc) and the light chain $\beta_2$-microglobulin ($\beta_2$m). Encoded by the most polymorphic gene cluster in humans with more than 24,000 known classical human leukocyte antigen (HLA)-A, -B, and -C alleles, all heavy chains invariantly share their association with $\beta_2$m. Prior to presenting peptides on the cell surface, MHC I molecules undergo a stringent quality control during their maturation process in the endoplasmic reticulum (ER) by transiently interacting with specialized proteins, including glycan sensors, disulfide isomerases, and specific MHC I chaperones[3]. These ER chaperone systems play a pivotal role in adaptive immunity not only by stabilizing folding intermediates but also by shaping the peptide repertoire displayed on MHC I molecules.

The two MHC I-specific chaperones, tapasin and TAP-binding protein related (TAPBPR), were shown to share similar binding interfaces on MHC I and were thus suggested to have common catalytic

principles[4–8]. However, the two chaperones differ substantially in their subcellular localization and their molecular liaisons. TAPBPR acts autonomously on a subset of MHC I clients in the peptide-depleted environments of the *cis*-Golgi and ER-Golgi intermediate compartment (ERGIC)[9]. As the major MHC I editor, tapasin fulfils its function predominantly as part of the peptide-loading complex (PLC) in conjunction with the oxidoreductase ERp57 and the lectin-like chaperone calreticulin in the ER[3,8]. The interaction between tapasin and ERp57 relies on a mixed disulfide between $Cys^{95}$ of tapasin and $Cys^{33}$ of ERp57 ($Cys^{57}$ in immature ERp57)[4]. In the oxidizing environment of the ER, $Cys^{33}$ of ERp57 toggles between an intermolecular, mixed disulfide with $Cys^{95}$ of tapasin and an intramolecular disulfide with $Cys^{36}$ ($Cys^{60}$ in immature ERp57)[10]. This toggling, known as the escape pathway, has been described for protein-disulfide isomerase-assisted protein folding and prevents the enzyme from becoming trapped in covalent complexes with its substrates[11].

Peptide-deficient MHC I molecules are recruited to the PLC[12,13], undergo peptide loading and editing[14,15], dissociate from the PLC as stable peptide–MHC I (pMHC I) complexes, and traffic to the cell surface for antigen presentation[3]. During peptide loading and editing, tapasin stabilizes peptide-receptive MHC I heterodimers

[1]Institute of Biochemistry, Biocenter, Goethe University Frankfurt, Frankfurt/Main, Germany. [2]Department of Immunopathology, Sanquin Research, Amsterdam, The Netherlands. [3]Landsteiner Laboratory, Amsterdam UMC, University of Amsterdam, Amsterdam, The Netherlands. ✉e-mail: trowitzsch@biochem.uni-frankfurt.de; tampe@em.uni-frankfurt.de

and catalyzes peptide exchange by accelerating peptide association and dissociation[15–17]. Deletion of tapasin and ERp57 results in an impaired immune response and altered peptide repertoire[18,19]. The various MHC I allomorphs exhibit distinct levels of plasticity, differ in peptide-binding preferences, and depend to different degrees on tapasin for optimal peptide loading[20,21]. Conformational plasticity of MHC I scales with the ability to select high-affinity peptides in the absence of tapasin[22–25]. Insights into the molecular basis of tapasin promiscuity for different MHC I clients as well as structural elements necessary for peptide exchange catalysis at atomic resolution are lacking due to the intrinsically transient nature of the editing complex.

Here, we present the 2.7-Å crystal structure of an MHC I–tapasin–ERp57 editing complex assembled by a photo-triggered approach. The structure of this transient assembly reveals molecular rearrangements within the chaperone complex upon client engagement. Movement of the C-terminal immunoglobulin domain of tapasin extends the interaction interface to MHC I hc and $\beta_2$m. We show by functional assays that structure-guided ablation of interface interactions leads to reduced cell surface presentation of pMHC I complexes. The editing loop of tapasin rigidifies upon client recognition and contributes to a widened MHC I F-pocket. Since the loop structure might also stabilize a peptide-bound MHC I pocket, our data suggest a dual function of the editing loop in peptide exchange catalysis. Our structural and functional data reveal the molecular basis of tapasin promiscuity towards MHC I clients and suggest that the level of plasticity of peptide-receptive MHC I determines chaperone-assisted acquisition of high-affinity peptides.

## Results and discussion

### Structure of the MHC I–tapasin–ERp57 editing complex

We captured a human tapasin–ERp57 heterodimer with bound MHC I hc and $\beta_2$m *in flagrante* by a photo-triggered approach[7] and determined the crystal structure of the MHC I-chaperone complex to 2.7 Å resolution (Fig. 1). To facilitate crystallization, we utilized the ectodomains of tapasin and MHC I hc lacking the transmembrane and cytosolic regions. After photo-cleavage of loaded peptides, we first screened different MHC I allomorphs for complex formation with the tapasin–ERp57 heterodimer by size exclusion chromatography (SEC), resulting in murine H2-D$^b$ as the most suited MHC I hc candidate (Fig. 1a, b, and Supplementary Fig. 1a–d). Furthermore, we substituted Cys$^{36}$ of ERp57 with Ala (C36A), formerly described as C60A in immature ERp57 (ref. 4), to retain the mixed disulfide between tapasin and ERp57 (ref. 10).

Since a homogeneous population of a kinetically stable chaperone-client complex between disulfide-linked tapasin–ERp57 and refolded MHC I hc/$\beta_2$m could not be isolated (Fig. 1a, b), we directly used a photoactivated mix of refolded photo-peptide/H2-D$^b$/$\beta_2$m and tapasin–ERp57 (1.3: 1 molar ratio) for crystallization. Size exclusion chromatography-coupled mass spectrometry (SEC-MS) revealed an almost complete photo-induced cleavage (>96%) of the peptide bound to MHC I (Supplementary Fig. 2). Initial crystals of the

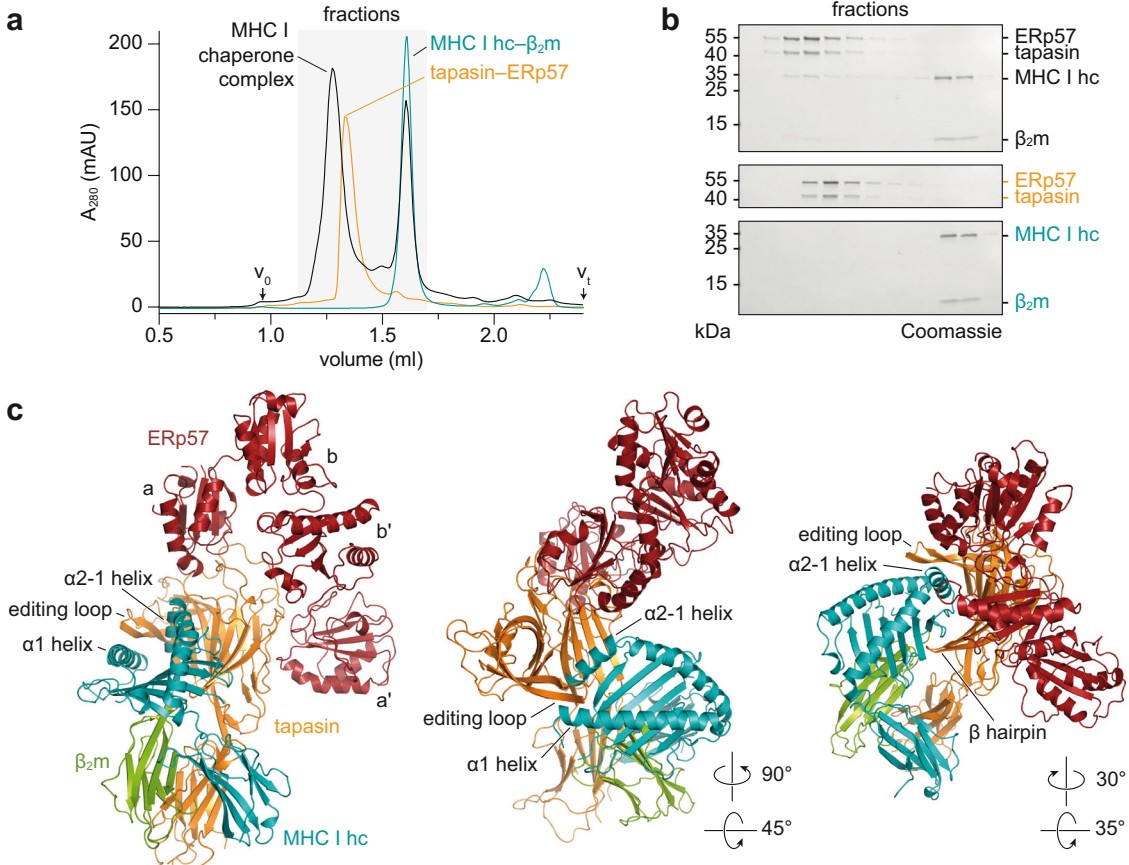

**Fig. 1 | Photo-triggered assembly and structural overview of the MHC I–tapasin–ERp57 complex. a** Photo-triggered assembly of a transient MHC I–tapasin–ERp57 complex (black) analyzed by size exclusion chromatography (SEC). SEC elution profiles of tapasin–ERp57 and MHC I hc–$\beta_2$m are colored in orange and teal, respectively. $\beta_2$m $\beta_2$-microglobulin; hc heavy chain; $A_{280}$ absorption at 280 nm; $V_0$ void volume; $V_t$ total volume. **b** MHC I–tapasin–ERp57 complex formation monitored by SEC and SDS-PAGE. kDa kilodalton; *n* = 1. Same color code as illustrated in **a** is used. **c** Cartoon representation of peptide-receptive MHC I (MHC I hc, teal; $\beta_2$m, green) in complex with the ER chaperone tapasin–ERp57 (tapasin, orange; ERp57, red) in different orientations. a, b, b', and a', thioredoxin-like domains of ERp57. Source data for **a** and (**b**) are provided as a Source Data file.

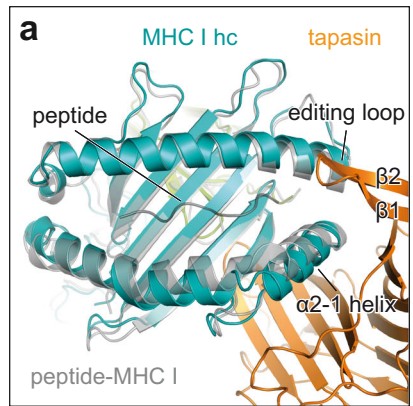

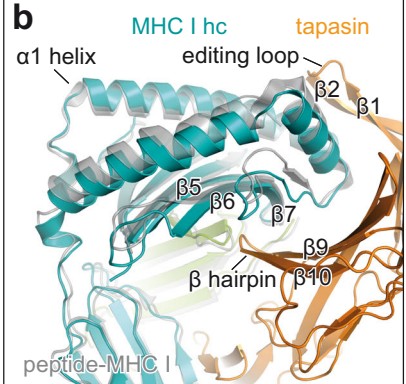

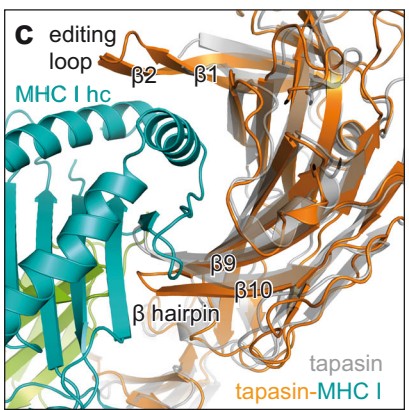

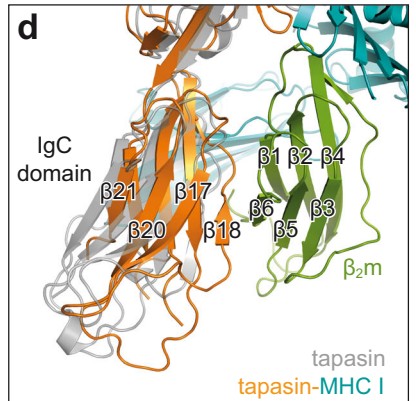

**Fig. 2 | Conformational changes upon client-chaperone engagement. a** Top view of superpositing peptide-receptive MHC I in the chaperone complex (MHC I hc, teal; tapasin, orange) with peptide-bound H2-D$^b$ (gray, PDB ID 2F74) in cartoon representation. hc heavy chain; β1, β2, β strands of editing loop of tapasin. **b** Side view of the superposition shown in **a** β5, β6, β7, β strands of MHC I hc; β9, β10, β hairpin of tapasin. **c** View onto the concave interface of tapasin in superposition of MHC I-bound tapasin from the chaperone complex (tapasin, orange; MHC I hc, teal; β$_2$m, green) with client-free tapasin (gray, PDB ID 3F8U). **d** Superposition as in **c** but viewed onto the C-terminal domain of tapasin. IgC domain immunoglobin constant domain, β$_2$m β$_2$-microglobulin.

chaperone-client complex belonging to space group P2$_1$2$_1$2$_1$ diffracted X-rays to 4.0 Å resolution with two heterotetrameric editing complexes in the asymmetric unit. Utilizing microseed matrix screening[26], we obtained crystals in space group P22$_1$2$_1$ with an improved resolution of 2.7 Å and only one complex in the asymmetric unit, enabling us to build an atomic model with good stereochemistry (Fig. 1c and Supplementary Table 1).

As in the client-free structure of tapasin–ERp57 (ref. 4), the four thioredoxin-like domains of ERp57 (a-b-b′-a′) are arranged in their characteristic twisted U form, whereas tapasin forms its typical L-shaped structure[4,8] (Fig. 1c and Supplementary Movie 1). The a and a′ domains of ERp57 interact with the N-terminal region of tapasin composed of a fusion between a seven-stranded β barrel and an immunoglobin (Ig)-like fold[4]. In contrast to the structure of client-free tapasin–ERp57 (ref. 4), we could not model the mixed disulfide between Cys$^{95}$ of tapasin and Cys$^{33}$ of ERp57 present in the purified tapasin–ERp57 heterodimer (Supplementary Fig. 1). Since disulfide bond reduction did not result from UV illumination performed for photocleavage of the peptide (Supplementary Fig. 1), we speculate that reduction is either facilitated by X-ray irradiation[27] or resulted from the MHC I editing process itself.

## Molecular rearrangements within the chaperone complex upon client engagement

The interface between tapasin and MHC I covers a total surface area of 4381 Å$^2$ (hc: 74%; β$_2$m: 26%) (Supplementary Fig. 3) of which the N-terminal domain of tapasin contributes 2657 Å$^2$. The concave binding site of the N-terminal domain embeds the α2-1 helix region of MHC I hc like a clamp (Fig. 1c). On one side of the clamp, a loop comprising

residues Glu$^{11}$-Lys$^{20}$, referred to as editing loop in the following, sits on top of the F-pocket of the MHC I peptide-binding groove, whereas a β hairpin element contacts the floor of the peptide-binding groove from below (Fig. 2a–c). When superimposed with atomic models of client-free tapasin–ERp57 (ref. 4) and peptide-bound MHC I (ref. 28), our structure reveals that client binding leads to important rearrangements of the α1 and α2-1 helices in MHC I, resulting in a widened F-pocket (Fig. 2a). This widening is stabilized by a hydrogen-bond network between His$^{70}$ and Glu$^{72}$ of tapasin, which attracts the side chain of Tyr$^{84}$ of the MHC I hc (Fig. 3a, b). The side chain of Lys$^{20}$ of tapasin thereby facilitates positioning of Tyr$^{84}$ by hydrogen bonding with the main-chain oxygen (Fig. 3b). Notably, Glu$^{72}$ of tapasin structurally corresponds to Glu$^{105}$ of TAPBPR, which was found to contact Tyr$^{84}$ of MHC I hc in a similar manner[6,7]. The β-sheet floor (β-strands β5-7) of the peptide-binding groove near the F-pocket is distorted upon binding, presumably by engagement with the β hairpin of tapasin (β strands β9-10) (Fig. 2b). Upon client binding, the β hairpin of tapasin is shifted downwards to accommodate the α2-1 region of MHC I (Fig. 2c) in a similar way as revealed in TAPBPR–MHC I structures[6,7]. Furthermore, the C-terminal Ig-like domain of tapasin swings towards the α3 domain of the MHC I hc and β$_2$m (Fig. 2d), allowing for an additional interaction interface with a buried surface area of 1724 Å$^2$ (hc: 55%; β$_2$m: 45%; Supplementary Fig. 3a, b).

## The editing loop of tapasin contributes to a widened MHC I F-pocket

Upon client binding, the intrinsically disordered editing loop of the client-free state of tapasin[4,29] rigidifies as evidenced by a clear electron density at the 1.5 σ level for most loop residues (Fig. 3a–d). The editing

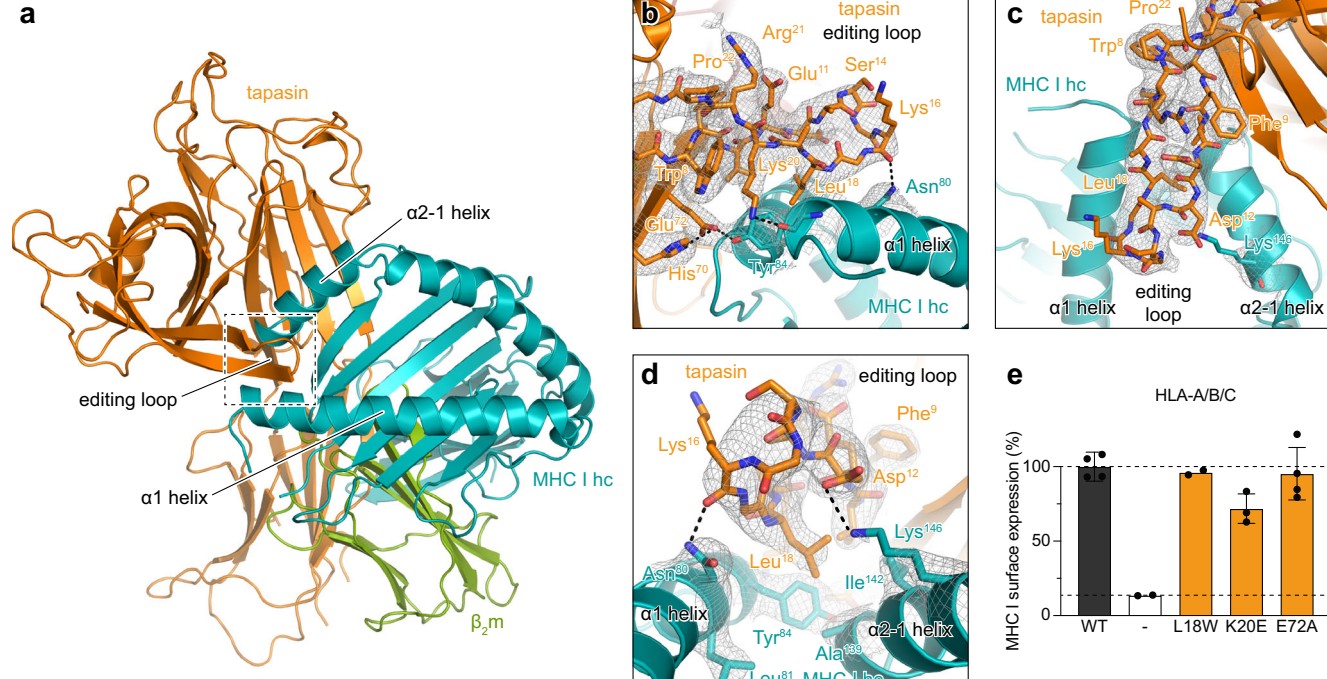

**Fig. 3 | The editing loop of tapasin contributes to a widened F-pocket of MHC I.**
**a** Cartoon representation of the chaperone complex as shown in Fig. 1c. Tapasin, MHC I hc and $\beta_2$m are colored in orange, teal and green, respectively. Region magnified in **b**–**d** is indicated by a dashed box. ERp57 is not shown for the sake of clarity. **b** Side view onto the editing loop (stick representation) of tapasin with corresponding electron density map (contour level: 1.5 σ). Dashed lines represent hydrogen bonds or salt bridges. The same color code as in **a** is applied. hc heavy chain. **c** Top view onto the editing loop with corresponding electron density and color code as shown in **b**. **d** Close-up view into the F-pocket of MHC I with corresponding electron density and color code as displayed in **b**. **e** MHC I surface expression of tapasin-deficient HAP1 cells (white), expressing wildtype (WT, dark

gray) or different interaction mutants of tapasin (orange). Flow cytometry was performed using an APC-conjugated pan-HLA-A/B/C-specific antibody (W6/32). The lower black dashed line represents the level of MHC I surface expression of cells transfected with a vector devoid of gene of interest. The upper black dashed line represents the level of MHC I surface expression of cells expressing wildtype tapasin. The mean fluorescence intensity of MHC I surface expression was normalized to wildtype tapasin (±SD; n = 4 biologically independent samples; K20E, n = 3 biologically independent samples; L18W and mock transfected, n = 2 biologically independent samples). The gating strategy is displayed in Supplementary Fig. 8. WT wildtype, − mock transfected. Source data for **e** are provided as a Source Data file.

loop is stabilized by the side chain of Asn[80] of MHC I, which forms a hydrogen bond with the backbone-carbonyl oxygen of Lys[16] of tapasin (Fig. 3d). The side chain of Asp[12] of tapasin is in salt-bridge distance to the ε-amino group of Lys[146] of the MHC I hc that presumably contributes to the widening of the peptide-binding groove and the rigidification of the editing loop of tapasin (Fig. 3d). Furthermore, the side chain of Leu[18] of the editing loop, which was shown to be relevant for MHC I surface expression[30,31], compensates in the peptide-free MHC I F-pocket for the lack of a C-terminal hydrophobic side chain of the antigenic peptide and presumably assists in groove widening. Even though sitting directly above the F-pocket, Leu[18] does not mimic the C terminus of cargo peptides. Thus, Leu[18] is unlikely to act as a peptide surrogate per se[30]. In line with this groove-widening function, substitution of Leu[18] with a bulky hydrophobic side chain (tryptophan) did not negatively impact MHC I surface presentation in our flow cytometry-based cellular assays when compared to wildtype (WT) (Fig. 3e and Supplementary Fig. 3c). Conversely, substitution of Leu[18] with glycine has been shown to nearly abolish peptide exchange activity of tapasin[31].

Surprisingly, abrogation of the hydrogen bond between Tyr[84] of MHC I and Glu[72] of tapasin did not affect MHC I surface presentation (Fig. 3e), as demonstrated by an in vitro peptide-loading assay[4]. Introducing a charge repulsion by mutating tapasin Lys[20], which helps in positioning the α1 helix of MHC I via hydrogen bonding to the backbone-carbonyl oxygen of Tyr[84] (Fig. 3a, b), did not substantially decrease the surface MHC I level (72 ± 10% of WT; Fig. 3e), suggesting that the editing loop of tapasin mainly stabilizes the empty F-pocket by a hydrophobic plug. At the same time, MHC I F-pocket residues Asn[80],

Tyr[84], and Lys[146], which coordinate the terminal carboxyl group of bound peptides, are engaged in interactions with tapasin and its editing loop (Fig. 3a, b and Supplementary Fig. 4a). Our data suggest that tapasin acts in a dual manner, opening the peptide-binding groove and stabilizing the empty F-pocket of the MHC I hc.

## Domain movements in tapasin extend the interaction interface with MHC I

The interface loop region of tapasin on the lateral side of the MHC I α2-1 helix[4,7] is not involved in specific interactions and therefore only plays a minor role in MHC I surface presentation (Fig. 4a). In contrast, the neighboring Gln[261] of tapasin, which is involved in several hydrogen bonds with the α2-1 helix region, leads to a reduction of MHC I surface presentation to 65 ± 6% when mutated to alanine (Fig. 4b, f and Supplementary Fig. 3d). An even stronger suppression of surface presentation was observed when interactions between the β hairpin of tapasin and the α2 domain of MHC I hc were disrupted (R187E; 55 ± 13% compared to WT) (Fig. 4c, f). Mutations in both the interface loop and the β hairpin synergistically reduced MHC I surface presentation to 46 ± 4% (Fig. 4f). Interestingly, as in free tapasin–ERp57 (ref. 4), we observed an intermolecular contact between side chains of Arg[333] of tapasin and Glu[229] of MHC I hc (Fig. 4d), which, when disturbed by charge repulsion, reduced MHC I surface presentation to 37 ± 3% (Fig. 4f). The importance of this interaction between the CD8-recognition loop of MHC I and the IgC domain of tapasin was already proposed by the cryogenic electron microscopy structure of the PLC[8]. Disrupting the hydrogen bond between Ser[336] of tapasin and Thr[225] of MHC I hc distant from the β-sheet interaction[32] had almost no

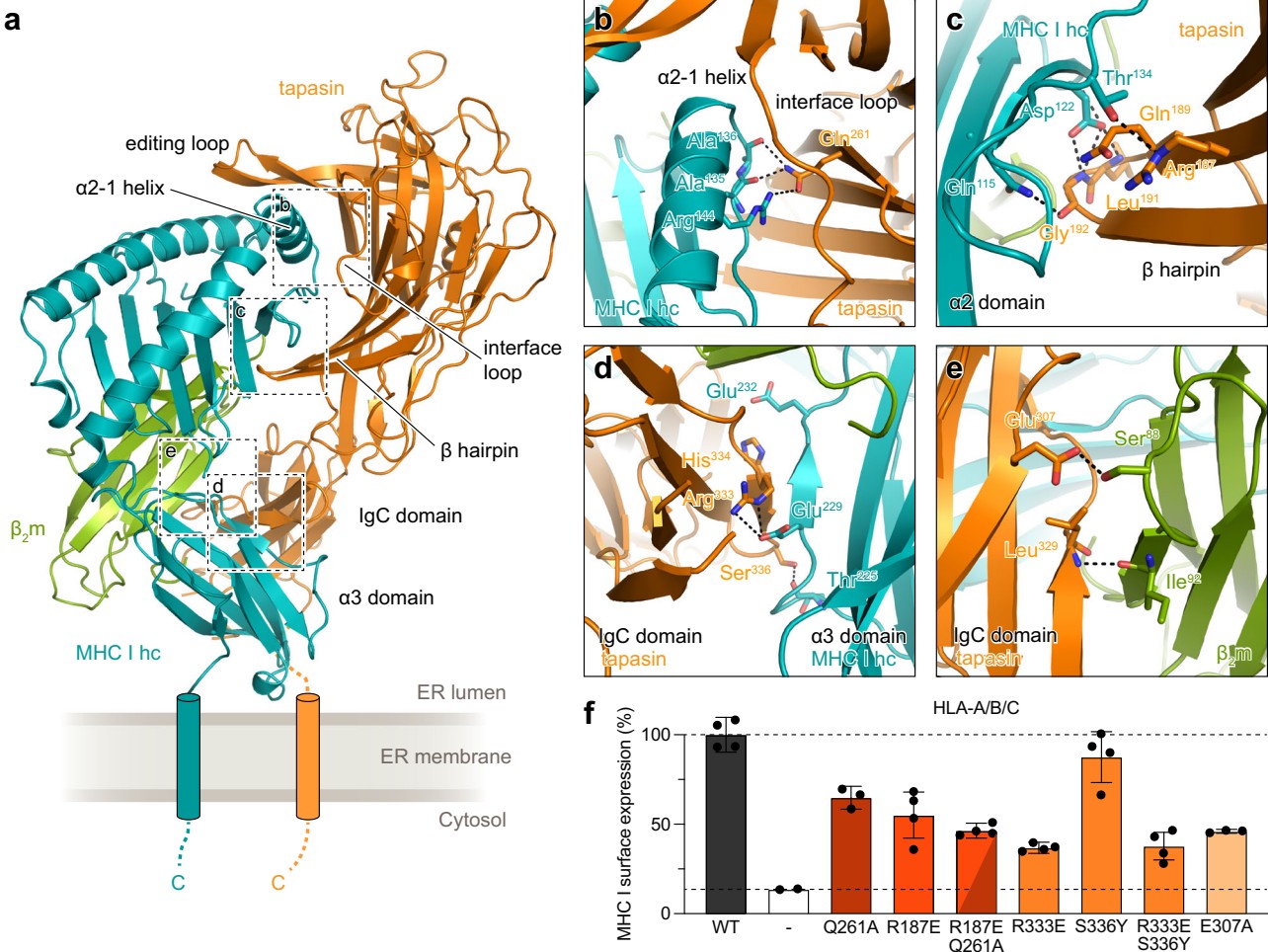

**Fig. 4 | Molecular interactions at chaperone-client interfaces affect MHC I surface expression. a** Cartoon representation of the chaperone complex as shown in Fig. 1c. The components tapasin, MHC I hc and β₂m are displayed in orange, teal and green, respectively. Regions magnified in **b**−**e** are indicated by dashed boxes. ERp57 is not shown for the sake of clarity. Transmembrane helices and C-terminal regions (C) of MHC I hc and tapasin are indicated schematically. **b** Zoom-in of the α2-1 helix region of MHC I and tapasin. Interface residues forming hydrogen bonds (black dashed lines) are shown as sticks. **c** Magnification of the interface formed between the β hairpin of tapasin and the α1/α2 domains of MHC I. **d** Close-up view of the C-terminal domain of tapasin and the α3 domain of MHC I. **e** Interface formed between β₂m and the IgC domain of tapasin. **f** Restored MHC I surface expression of tapasin-deficient HAP1 cells by interface mutants of tapasin as assessed by flow cytometry. Mean fluorescence intensities of cells stained with APC-conjugated pan-HLA-A/B/C-specific antibody (W6/32) were normalized to the tapasin amount of the wildtype (dark gray, upper dashed line) (±SD, $n = 4$ biologically independent samples; Q261A, R333E/S336Y, and E307A, $n = 3$ biologically independent samples; mock transfected, $n = 2$ biologically independent samples). The lower dashed line represents the level of MHC I surface expression of mock-transfected cells (white). The gating strategy is displayed in Supplementary Fig. 8. α2-1 interface, brown; β-hairpin interface, dark orange; α3 interface, orange; β₂m interface, light orange. WT wildtype, − mock transfected. Source data for **e** are provided as a Source Data file. The same color code as used in **a** is applied in **b**−**e**.

impact on surface presentation (Fig. 4d, f). The displaced IgC domain of tapasin is further stabilized by contacts to β₂m, involving backbone interactions between Ile⁹² of β₂m and Leu³²⁹ of tapasin and a side chain hydrogen bond between Ser⁸⁸ of β₂m and Glu³⁰⁷ of tapasin (Fig. 4e). Disruption of the hydrogen bond between Glu³⁰⁷ of tapasin and Ser⁸⁸ of β₂m reduced the MHC I surface level to 46 ± 1% (Fig. 4e, f); albeit the expression level of mutant tapasin was also lowered compared to WT (Supplementary Fig. 3e).

## Dual function of the editing loop in peptide exchange catalysis

Mechanistic studies suggested that the MHC I-specific chaperones tapasin and TAPBPR hold distinct preferences for MHC I allomorphs[33–35]. However, our structure of the MHC I−tapasin−ERp57 editing complex demonstrates that the binding interfaces used for client recognition are similar between the two editors with minor differences for specific residues (Supplementary Figs. 5 and 6). Comparing the impact of the two editors on widening the peptide-binding cleft reveals a slightly stronger displacement of the α2-1 helix of MHC I when associated with TAPBPR (Supplementary Fig. 4b). In our structure, the editing loop lies on top of the MHC I F-pocket (Fig. 3). In chaperone-free pMHC I, the side chains of residues Asn⁸⁰ and Lys¹⁴⁶ point into the F-pocket and coordinate the C terminus of bound peptides (Supplementary Fig. 4a). In an intermediate peptide-/chaperone-bound state the carbonyl group of Gly¹⁵ of the editing loop may present a hydrogen-bond acceptor, which could shield the peptide-bound MHC I F-pocket, consistent with a peptide trapping function[29]. We propose that the editing loop fulfills two putatively contrasting tasks. On the one hand, the editing loop contributes to the widening of the F-pocket in the peptide-receptive state of MHC I, whereas, on the other hand, it shields the peptide C terminus during the epitope-loading process. Such a scenario may also be applicable to TAPBPR as NMR studies showed that the longer editing loop of TAPBPR (residues Gly²⁴-Arg³⁶) can be observed as a lid on top of the F-pocket[29], whereas in one of the MHC I/TAPBPR structures, the loop dives into the F-pocket and stabilizes peptide-receptive MHC I by acting as a peptide surrogate (scoop loop)[7,36,37] (Supplementary Fig. 4b).

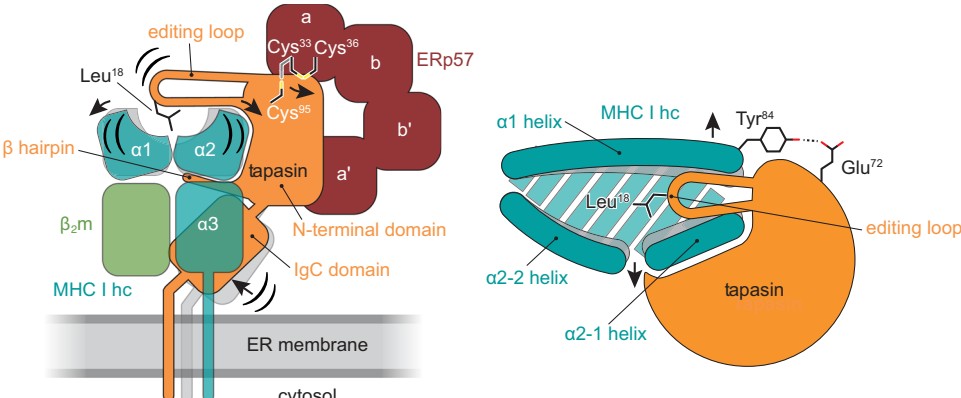

**Fig. 5 | Model for chaperone function of tapasin towards peptide-receptive MHC I.** Schematic of molecular motions leading to stabilized peptide-receptive clients (MHC I hc, teal; $\beta_2$m, green) upon association with tapasin (orange; left, side view; right, top view). Conformational changes are indicated by black arrows. The dotted line indicates a polar interaction between tapasin (Glu[72]) and chaperoned MHC I (Tyr[84] of the heavy chain, hc). For reasons of clarity, ERp57 (red) is not shown in the right panel.

## Tapasin promiscuity for MHC I clients

In the tapasin-chaperoned state, the position of $\beta_2$m in relation to the MHC I hc is similar (RMSD 1.3 Å) to the peptide-bound state[28], whereas binding of TAPBPR leads to a repositioning of $\beta_2$m[6,7]. Strikingly, sequence alignments of classical MHC I allomorphs show that the interaction sites for tapasin are conserved among tapasin-dependent and -independent clients and are shared between tapasin and TAPBPR, except for residue Trp[73] of H2-D[b] (Supplementary Fig. 6). Analyzing the primary structure of non-classical MHC I molecules HLA-E, -F, and -G demonstrated that not all potential interacting residues are conserved, but most of the alternative residues are still capable to form hydrogen bonds (Supplementary Fig. 7). In essence, the crystal structure of the MHC I–tapasin–ERp57 editing complex reveals that both tapasin-dependent and -independent allomorphs have the structural pre-requisites to interact with the chaperone, suggesting that their degree of plasticity and hence their ability to accommodate high-affinity peptide cargo determines the association, which is stabilized by key structural elements of tapasin (Fig. 5). It needs further structural and functional characterization to understand how e.g., C-terminally extended peptides that protrude from the MHC I F-pocket are edited by tapasin or TAPBPR. As tapasin acts primarily as part of the PLC multichaperone network, we speculate that association of ERp57 with tapasin is not only necessary for coordinating the glycan-chaperone calreticulin, but that the molecular dynamics of this interaction may help unlock tapasin for client engagement and peptide proofreading catalysis.

## Methods
### Plasmid constructs
The constructs of human $\beta_2$-microglobulin ($\beta_2$m) (UniProtKB P61769; amino acids 21-119) and of the ectodomain of H2-D[b] (UniProtKB P01899; amino acids 21-300) were previously described[7]. DNA fragments encoding the ER-lumenal domain of human tapasin (UniProtKB O15533; amino acids 21-402; tapasin[ΔTMD]) and human ERp57 (UniProtKB P30101; amino acids 25-505) were amplified by polymerase chain reaction (PCR) and cloned into the modified MultiBac vectors pAMI and pMIDK, respectively[38]. The C36A escape pathway mutation (ERp57[C36A]) was introduced in the ERp57-encoding construct by Quik-Change mutagenesis[11]. A C-terminal His$_6$-tag and a Tobacco Etch Virus (TEV) protease cleavage site were introduced into the pAMI vector encoding tapasin[ΔTMD] by sequence- and ligation-independent cloning (SLIC)[39]. The pMIDK_ERp57[C36A] donor vector was fused by Cre-mediated recombination to the pAMI_tapasin[ΔTMD] acceptor vector, resulting in one transfer plasmid for integration into the EMBacY

baculoviral genome via Tn7 transposition[40]. Tapasin-encoding plasmids for flow cytometry analyses were generated by assembling the coding sequence of wildtype tapasin (UniProtKB O15533; amino acids 21-448), an internal ribosomal entry site 2 (IRES2) followed by the enhanced green fluorescent protein (eGFP)-encoding sequence into the pAMI vector. Interaction mutations of tapasin were introduced by SLIC[39]. All vectors were verified by Sanger sequencing.

### Protein production and purification
H2-D[b] and $\beta_2$m were expressed as inclusion bodies in *Escherichia coli* Rosetta(DE3)pLysS as described[41]. In brief, Rosetta(DE3)pLysS cells were transformed with plasmids encoding $\beta_2$m or H2-D[b]. Overnight cultures of 20-30 ml in TYP media (16 g tryptone, 16 g yeast extract, 5 g NaCl, 1.25 g K$_2$HPO$_4$ per 1 l) supplemented with ampicillin and chlor-amphenicol were used to inoculate 2 l of TYP main culture. The main culture was incubated at 37 °C and 160 rpm until an optical density at 600 nm of 0.8 was reached. Temperature was reduced to 20 °C and protein production was induced by addition of 1 mM isopropyl-β-D-thiogalactopyranoside (IPTG). After 20 h, cells were harvested by centrifugation at 4 °C and 4500 × g for 15 min. Cells were flash-frozen in liquid nitrogen and stored at −80 °C.

The tapasin–ERp57 heterodimer was overexpressed in *Spodoptera frugiperda* (*Sf*) 21 cells essentially as previously described[40,42]. In brief, *Sf*21 cells were grown in Sf900 II SFM medium (Thermo Fisher Scientific) at 28 °C and transfected with modified EMBacY bacterial artificial chromosomes (BACs) using X-tremeGENE DNA transfection reagent (Roche). After incubation for 72 h at 28 °C, recombinant baculovirus V$_0$ was harvested and utilized for production of amplified baculovirus V$_1$. The tapasin–ERp57 heterodimer was expressed in 800 ml suspension culture at a cell density of 10$^6$ cells ml$^{-1}$ by infection with 0.5–1.0% (v/v) baculovirus V$_1$. Cells were harvested 72 h post cell proliferation arrest by centrifugation at 4 °C and 1300 × g for 10 min. Cell pellets were flash-frozen in liquid nitrogen and stored at −80 °C.

### Peptide synthesis
The photocleavable peptide photo-P18-I10 (RGPGRAF(J*)TI; J*, (S)−3 amino-3-(2-nitrophenyl)-propanoic acid) was synthesized using Fmoc solid-phase chemistry on a Liberty Blue Automated Microwave Peptide Synthesizer (CEM Corporation). A Wang resin preloaded with iso-leucine (Iris Biotech GmbH) was utilized as support. After synthesis, peptides were cleaved from the Wang resin using 95% trifluoroacetic acid (TFA), 2.5% H$_2$O and 2.5% triisopropylsilane (TIPS). Peptides were washed four times with diethyl ether, resuspended in a mixture of tert-butanol and water (4:1), and lyophilized. Peptide identity was verified

on a liquid chromatography-mass spectrometer (BioAccord LC-MS, Waters). $M_{calc}$: 1166.599 Da; $M_{obs}$: 1166.601 Da.

## Refolding and assembly of H2-D$^b$/β$_2$m complexes

Inclusion bodies of β$_2$m and H2-D$^b$ were isolated and refolded by dialysis according to established protocols[41]. First, cell pellets were resuspended in TE buffer (20 mM Tris-HCl, pH 8.0, 100 mM NaCl, 10 mM DTT) containing 1% (v/v) Triton X-100, followed by centrifugation at $10,000 \times g$ for 20 min at 4 °C. Inclusion bodies were washed five times followed by two washing steps with TE buffer without Triton X-100. Pure inclusion bodies were resuspended in TE buffer, flash-frozen in liquid nitrogen, and stored at −80 °C in aliquots.

Purified inclusion bodies from a 2 l expression culture of β$_2$m were dissolved in 50 ml denaturation buffer (8 M urea, 100 mM Tris-HCl, pH 8.0) and incubated at RT for 1 h. After centrifugation at $16,000 \times g$ for 5 min at 4 °C, supernatant was dialyzed twice against 2 l of 10 mM Tris-HCl, pH 8.0, for at least 9 h. Refolded β$_2$m was polished by size exclusion chromatography (SEC) using a Superdex75 16/60 column (GE Healthcare) equilibrated with 50 mM HEPES-NaOH, pH 7.4, 150 mM NaCl. Elution fractions corresponding to β$_2$m were concentrated by ultrafiltration (Amicon Ultra 3 kDa, MWCO, Merck), flash-frozen in liquid nitrogen, and stored at −80 °C.

Inclusion bodies of H2-D$^b$ were isolated and refolded by rapid dialysis according to established protocols[7,41]. H2-D$^b$ inclusion bodies were dissolved in denaturation buffer (8 M urea, 100 mM Tris-HCl, pH 8.0) and added dropwise under constant stirring at 4 °C (final concentration of 1 mM) to refolding buffer (100 mM Tris-HCl, pH 8.0, 400 mM L-arginine, 5 mM reduced glutathione, 0.5 mM oxidized glutathione, 2 mM ethylenediaminetetraacetate (EDTA), 1x Protease-Inhibitor Mix HP (Serva, Electrophoresis GmbH)), in the presence of 40 mM photocleavable peptide photo-P18-I10 and 2 mM of purified β$_2$m. The refolding reaction was incubated under stirring for 3 days at 4 °C. Refolded peptide/H2-D$^b$/β$_2$m complexes were concentrated by ultrafiltration (Amicon Ultra 10 kDa, MWCO, Merck) and polished by SEC using a Superdex200 10/300 Increase column (GE Healthcare) equilibrated in SEC buffer (50 mM HEPES-NaOH, pH 7.4, 150 mM NaCl). Peak fractions of refolded photo-P18-I10/H2-D$^b$/β$_2$m complexes were pooled, concentrated by ultrafiltration, flash-frozen in liquid nitrogen, and stored at −80 °C.

## Purification of tapasin–ERp57 heterodimers

Infected *Sf*21 cell pellets were resuspended in 100 ml lysis buffer (50 mM HEPES-NaOH, pH 7.4, 150 mM NaCl, 25 mM imidazole, 1 mM phenylmethylsulfonyl fluoride (PMSF), 1 mM benzamidine) per 1 l expression culture and lysed by sonication. After centrifugation at $20,000 \times g$, 4 °C for 30 min, the supernatant was incubated with pre-equilibrated Ni$^{2+}$-nitrilotriacetic acid (NTA) agarose resin (Thermo Fisher Scientific) for 1 h at 4 °C. NTA agarose resin was washed with lysis buffer without protease inhibitors and bound proteins were eluted with elution buffer (50 mM HEPES-NaOH, pH 7.4, 150 mM NaCl, 300 mM imidazole). Protein-containing elution fractions were pooled, and buffer exchanged to 50 mM HEPES-NaOH, pH 7.4, 150 mM NaCl on a PD-10 desalting column (GE Healthcare). The His$_6$-tag was cleaved by addition of TEV protease and incubated overnight at 4 °C. Cleaved tapasin–ERp57 heterodimers were concentrated by ultrafiltration (Amicon Ultra 50 kDa, MWCO, Merck) and polished by SEC using a Superdex200 10/300 Increase column (GE Healthcare) equilibrated with 50 mM HEPES-NaOH, pH 7.4, 150 mM NaCl. Peak fractions were pooled, flash-frozen in liquid nitrogen, and stored at −80 °C.

## Crystallization

Purified tapasin–ERp57 and photo-P18-I10/H2-Db/β$_2$m complexes were mixed at a molar ratio of 1.0 to 1.3 at a total concentration of 10 mg ml$^{-1}$. Aliquots of 20 µl were exposed to UV light (365 nm, 185 mW/cm$^2$, 120 s) and applied to high-throughput crystallization

trials. Initial crystals belonging to the orthorhombic space group (SG) P2$_1$2$_1$2$_1$ with two tapasin–ERp57/H2-D$^b$/β$_2$m complexes per asymmetric unit were obtained at 18 °C by sitting drop vapor diffusion using a reservoir solution comprising 900 mM sodium cacodylate, pH 6.2, 270 mM sodium acetate, 6% polyethylene glycol (PEG) 2000, and 6% PEG500 monomethyl ether (MME). Despite extensive manual refinements and additive screening, these crystals diffracted X-rays only to a resolution of 4.0 Å. To obtain higher resolution, crystals were utilized for microseed matrix screening (MMS)[26]. To this end, crystals were crushed, transferred to a tube containing seed beads, and vortexed for 5 min at room temperature (RT). Seeds were used for another round of high-throughput crystallization trials. Crystals belonging to SG P22$_1$2$_1$ with one tapasin–ERp57/H2-D$^b$/β$_2$m complex per asymmetric unit were obtained by sitting drop vapor diffusion with a reservoir solution of 100 mM Gly-Gly, 100 mM 2-amino-2-methyl-1,3-propanediol (AMPD), pH 8.5, 300 mM lithium sulfate, 300 mM sodium sulfate, 300 mM potassium sulfate, 20% (v/v) PEG8000, 40% (v/v) 1,5-pentanediol, and a seed/protein/reservoir ratio of 0.3/1.0/1.0. Crystals were directly flash-frozen in liquid nitrogen without addition of further cryoprotectants.

## Data collection, structure determination, and refinement

The synchrotron data set was collected at beamline P13 operated by EMBL Hamburg at the PETRA III storage ring (DESY, Hamburg, Germany)[43]. Data were indexed, integrated, and scaled using the XDS package[44]. The structure of the editor complex was solved by molecular replacement using the coordinates of the tapasin–ERp57/MHC I editing module of the peptide-loading complex (PDB ID 6ENY)[8] as search model with the program Phaser within the Phenix software package[45]. The atomic model of the tapasin–ERp57/H2-D$^b$/β$_2$m complex was manually built in COOT[46] and refined in Phenix applying translation-liberation-screw (TLS) parameters. The atomic model was further improved using the PDB-REDO server[47] followed by manual adjustments in COOT[46] and refinement in Phenix[45]. The residual electron density in the MHC I binding pocket did not allow modeling of the photo-P18-I10 peptide used for refolding of MHC I or photo-induced fragments thereof with high occupancy. This result is consistent with SEC-MS analyses of H2-D$^b$/β$_2$m/photo-P18-I10 complexes after photo-triggered peptide cleavage that showed residual amounts of uncleaved peptide (4%) or of the 6-mer peptide fragment (2%) still bound to MHC I (Supplementary Fig. 2). Numberings of modeled amino acids correspond to the mature polypeptides lacking the respective signal sequences. Molecular graphics images were prepared using PyMOL (Schrödinger).

## Alignments

Multiple sequences were aligned using Clustal Omega[48] and plotted with ESPript 3.0 (ref. [49]).

## Cell culture

Tapasin-deficient HAP1 cell line[50] was cultured at 37 °C and 5% CO$_2$ in Iscove's modified Dulbecco's medium (IMDM, Gibco) supplemented with 10% fetal calf serum (FCS, Gibco). Cells were split at about 80% confluency with 0.05% Trypsin-EDTA (Gibco).

## Transfection of HAP1 cells

One day prior to transfection, $40 \times 10^3$ cells per well of a 6-well plate were seeded. On the day of transfection, cells were washed using 1× Dulbecco's phosphate buffered saline, pH 7.4 (DPBS, Gibco), and 1 ml of fresh IMDM medium per well was added. For the transfection mixture, 2 µg DNA and 6 µl X-tremeGENE HP (1:3 ratio) (Roche) were separately dissolved in 50 µl Opti-MEM (Gibco) and incubated for 5 min at RT. The solutions were mixed, incubated for 15 min at RT and added dropwise to the cells. 1 ml of medium was added after 4–6 h, and cells were harvested 48 h after transfection.

## MHC I surface expression

MHC I surface expression was analyzed by flow cytometry. All steps were carried out on ice. The cells were washed in 0.2 ml FACS buffer (1x DPBS, 2% BSA, 2 mM EDTA, 0.02% (w/v) sodium azide) and centrifuged for 5 min at $300 \times g$, 4 °C. The supernatant was discarded, and cells were blocked by 10% FcR Blocking Reagent, human (Miltenyi Biotec) in 50 µl FACS buffer for 15 min. Afterwards, cells were washed with 0.2 ml FACS buffer, centrifuged for 5 min at $300 \times g$ and 4 °C, and stained with 2 µl of APC anti-human HLA-A/B/C antibody (64 µg/ml, clone W6/32, BioLegend) in 0.1 ml FACS buffer for 30 min. Subsequently, cells were washed twice with 0.2 ml FACS buffer and resuspended in 0.2 ml FACS buffer for analysis by flow cytometry. Data was recorded on a FACSMelody Cell Sorter (BD Bioscience), processed using FlowJo V10 software and analyzed using Excel (Version 16.7) and GraphPad Prism 8.21 MacOS. The gating strategy is displayed in Supplementary Fig. 8.

## Immunoblotting

$0.2 \times 10^6$ eGFP-positive cells were sorted on FACSMelody Cell Sorter (BD Bioscience), centrifuged at $300 \times g$, 4 °C for 5 min, lysed in 25 µl Pierce RIPA buffer (Thermo Fisher Scientific) supplemented with 1% Benzonase (Novagen, EMD Chemicals) and 1× Protease-Inhibitor Mix HP (Serva, Electrophoresis GmbH), and incubated for 15 min at 500 rpm, 25 °C. 5 µl of 4× sodium dodecyl sulfate (SDS)-loading buffer (250 mM Tris-HCl, pH 7.5, 30% glycerol, 10% SDS, 10% β-mercaptoethanol, 0.02% Bromophenol Blue) were added, and samples were incubated at 95 °C for 10 min. 10 µl of samples were loaded on a Mini-PROTEAN TGX Gel (4-20%, BioRad), and gel electrophoresis was performed according to manufacturer's protocol (BioRad). The samples were blotted onto a PVDF membrane by semi-dry blotting for 30 min at 25 V. The blotted membranes were blocked in 5% milk in TBS-T for 1 h and incubated with Direct-Blot HRP anti-GAPDH (clone FF26A/F9, BioLegend, 1:2000) at 4 °C overnight. Blots were washed in TBS-T at RT for 20 min and then incubated with anti-tapasin (hybridoma, clone 7F6, 1:3000) for 1 h[51]. Anti-rat IgG Peroxidase conjugate (Sigma-Aldrich, 1:20,000) was incubated for 45 min at RT as secondary antibody. Membranes were incubated with Clarity Western ECL reagent (BioRad) or LumiGLO Peroxidase Chemiluminescent Substrate Kit (Seracare) and chemiluminescence was measured with a Fusion FX (Vilber). Due to weak intensities corresponding to GAPDH, membranes were incubated with unconjugated anti-GAPDH (PA1-987, Thermo Fisher Scientific, 1:2,000) at 4 °C overnight before incubation for 45 min with anti-rabbit IgG Peroxidase conjugate (Merck Millipore, 1:20,000).

## EndoH digest

For LC-MS analysis, tapasin–ERp57 was deglycosylated using EndoH. Glycoprotein was mixed with Glycobuffer 3 (New England Biolabs), and 625 U of EndoH (New England Biolabs) per 1 µg of protein were added. The mixture was incubated overnight at RT. Deglycosylated tapasin–ERp57 was purified by SEC.

## LC-MS analysis

All LC-MS measurements were performed using a BioAccord System (Waters). Peptides were analyzed on an ACQUITY UPLC Peptide BEH $C_{18}$ Column, 130 Å, 1.7 µm, 2.1 mm × 100 mm (Waters), with a linear water/acetonitrile gradient complemented with 0.1% (v/v) formic acid at 60 °C, 30 V cone voltage, 0.8 kV capillary voltage, and a desolvation temperature of 550 °C in positive polarity at 50–2000 m/z. Intact protein LC-MS data was acquired using a cone voltage of 60 V, 1.5 kV capillary voltage, and a desolvation temperature of 500 °C on an ACQUITY UPLC Protein BEH $C_4$ Column, 300 Å, 1.7 µm, 2.1 mm × 50 mm (Waters) at 80 °C running a linear water/acetonitrile gradient, supplemented with 0.1% (v/v) formic acid. Mass spectra were recorded in positive polarity at 2 Hz in full scan mode at 400–7000 m/z. SEC-MS measurements were performed using an ACQUITY UPLC Protein BEH SEC Column, 200 Å, 1.7 µm, 2.1 mm × 150 mm (Waters) in 20 mM ammonium acetate. Mass spectra were recorded in positive mode at 1 Hz in full scan mode at 400–7000 m/z with a cone voltage of 42 V, capillary voltage of 1.5 kV, and a desolvation temperature of 450 °C. Masses of peptides and proteins were calculated and confirmed in Unify. Intact mass spectra were deconvoluted in Unify using MaxEnt1 algorithm iterating to convergence. Spectra with high background noise were subjected to automatic baseline correction before deconvolution. Deconvoluted spectra were centroidized based on peak height and used for mass calculations. UV spectra were recorded at 280 nm with a sampling rate of 10 Hz.

## Reporting summary

Further information on research design is available in the Nature Research Reporting Summary linked to this article.

## Data availability

The structure of the MHC I–tapasin–ERp57 editing complex was deposited to the Protein Data Bank (http://www.rcsb.org) under accession number PDB ID 7QNG. Previously published structural data used in this study are accessible at the PDB (human PLC editing module tapasin-ERp57/calreticulin/MHC I, PDB ID 6ENY; peptide-receptive MHC I, PDB ID 2F74; client-free tapasin–ERp57 heterodimer, PDB ID 3F8U; MHC I–TAPBPR complex, PDB ID 5OPI). Mass spectrometry data are available via Zenodo [https://doi.org/10.5281/zenodo.5939241]. Source data are provided with this paper.

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

## Acknowledgements

The synchrotron MX data were collected at beamline P13 operated by EMBL Hamburg at the PETRA III storage ring (DESY, Hamburg, Germany). We thank Gleb Bourenkov for the assistance in using the beamline. We acknowledge Lukas Sušac for helpful comments on the paper, and Inga Nold and Andrea Pott for editorial work. This work was supported by the

German Research Foundation (GRK 1986/B4—Complex Light Control, TA157/12-1—Reinhart Koselleck Project, and CRC 1507—Membrane Assemblies, Machineries and Supercomplexes to R.T.) and the European Research Council (ERC Advanced Grant No. 789121 to R.T.).

## Author contributions

Conceptualization: S.T., R.T.; Methodology: I.K.M., C.W., C.T., S.T., R.T.; Investigation: I.K.M., C.W., S.T.; Knockout cells: R.M.S.; Visualization: I.K.M., S.T., R.T.; Funding acquisition: R.T.; Supervision: S.T., R.T.; Writing—original draft: I.K.M., S.T., R.T.; Writing—review and editing: I.K.M., C.W., C.T., R.MS., S.T., R.T.

## Funding

## Competing interests

The authors declare no competing interests.
