## [Peer Review File · Nature Communications]

Structure of an MHC I–tapasin–ERp57 editing complex defines chaperone promiscuityREVIEWER COMMENTS

Reviewer #1 (Remarks to the Author):

The structure of an MHC class I complex with tapasin is important for the field, even though there are few surprises from this structure, given the recent structures of the related complexes of MHC class I with TAPBPR (Jiang et al 2017 and Thomas and Tampe, 2017). Nonetheless, it is great to see this structure.

Some comments & suggestions:

1. page 4 and page 7: Escape pathway-explain what this is.

2. page 5: "Furthermore, the side chain of Leu18 of the editor loop, which was shown to be essential for MHC I surface expression 28,29,30," It appears that the leucine is not important for some MHC-I allotypes including H2-Db (Hafstrand et al).

3. since a large part of the discussion relates to the "editing loop", it would be helpful in Figure 3 to have data that unambiguously demonstrates the functional relevance of this loop to H2-Db assembly.

4. Since the C-domain of tapasin forms a significant interface with b2m, it would be useful to have some mutagenesis studies focused on that region.

5. Page 6 "Because MHC I F-pocket residues Thr143, Lys146, and Trp147, which coordinate the C terminus of bound peptides, are not blocked by residues of the editor loop (Supplementary Fig. 3a), these data suggest that the editor loop does not act as a peptide surrogate as discussed for TAPBPR7,27,31,32." The authors are referring to many prior publications from the Tampe lab which have highlighted a competitor/peptide surrogate role for the scoop loop of TAPBPR in TAPBPR-mediated editing, which has been controversial in the literature. They should clarify whether they are suggesting the mechanism of function of the editing/scoop loop is different in TAPBPR vs tapasin, and if so why. Alternatively, are they suggesting that their prior findings should be more cautiously interpreted? The parallel discussions on page 7 should also be clarified.

6. Overall, the presentation of Figures 3 and 4 could be improved by highlighting the specific region of the full complex that is in focus for each of those figures. Figure 5 (left panel) is also suggesting a peptide competitor model for the editing loop, which could be modified.

Reviewer #2 (Remarks to the Author):

The chaperone tapasin in conjunction with the oxidoreductase ERp57 is crucial for MHC I assembly and for shaping the epitope repertoire for high immunogenicity. The manuscript by I. K. Müller *et al.* resolve a crystal structure of a tapasin–ERp57 heterodimer in complex with a peptide-receptive MHC I heterodimer at 2.7 Å resolution, representing a transient ER quality control complex. They show the editing loop of tapasin rigidifies upon client recognition and contributes to a widened MHC I F-pocket, and suggest elements indispensable for peptide proofreading. Overall, the structural and functional work gives new insight into the mechanistic basis for the selector function of tapasin and reveal the level of plasticity of peptide-receptive MHC I determines chaperone-assisted acquisition of high-affinity peptides. The following are comments and questions the authors need to address.

1. It is not clear how the authors assembled the MHC I–tapasin–ERp57 editing complex by a photo-triggered approach. Please add more detailed introduction on the photo-triggered approach, and how did they induce the formation of the transient complex? Did they resolve the bound peptide in the crystal structure? If not, please explain why the peptide was not captured.
2. The authors describe on page 4 that “We utilized the C60A substitution in ERp57 to prevent the disulfide cleavage via the escape pathway”. Please add introduction on the escape pathway, and also explain why used the C60A substitution in ERp57 to prevent the disulfide cleavage via the escape pathway.
3. Regarding the crystal structure, the clashscore appears relatively high, and the structure need to be further refined and improved.
4. On page 4, the authors describe that “Disulfide reduction does not result from UV illumination performed for photocleavage of the peptide but is rather facilitated by X-ray irradiation (Supplementary Fig. 1e,f). Therefore, we speculate that the escape pathway is utilized to switch between a non-editing and an editing conformation of the tapasin–ERp57 heterodimer”. Please explain why the disulfide reduction is facilitated by X-ray irradiation, and how did the authors deduce the speculation “that the escape pathway is utilized to switch between a non-editing and an editing conformation of the tapasin–ERp57 heterodimer”.
5. The authors mention on page 5 that “..... a loop comprising the residues Glu11-Lys20 (referred to as editing loop) sits on top of the F-pocket", however it was later on referred to as “editor loop”. Please clarify the loop should be referred to as “editing loop” or “editor loop”.
6. The authors are suggested to give a more complete explanation on Fig. 5 in the text.

Reviewer #3 (Remarks to the Author):

Muller and colleagues present here the structure of MHC-I in complex with tapasin and ERp57 that show the fine mechanism of peptide loading, a critical step for the immune sensing of microbial infection. The manuscript is well written, clear, and the data excellent.

The complex is only transient in the cell, so being able to crystallise the complex is a tour de force, and required the use of photo-triggered technique. The structure reveal 2 loops that are driving the key interaction, one scooping the antigen binding cleft to stabilise and facilitate peptide exchange, and one that ensure the stability like a “firm hand” underneath the MHC that will support the fold of the protein known to be unstable if empty. Previously, the structure of HLA-II in complex with the HLA-DM molecule showed how peptide can go in and out of the cleft in HLA-II without the all “empty” MHC-II falling apart, we finally understand how different molecules (Tapasin and ERp57) work in concert to stabilise and swap peptide in the cleft.

Thanks for the movie it's very helpful to visualise the interaction between the molecules.

Comments & questions

Page 2

According to the <https://www.ebi.ac.uk/ipd/imgt/hla/about/statistics/> web site there are 23417 HLA-A/B/C alleles identified so far, probably worth updating the number reported in the introduction.

Page4

What other MHC, beside H2Db, were screened and what was the difference with H2Db? Would the authors have an explanation for why H2Db was better for structural determination?

Page5

The total surface area between the tapasin and MHC-I is mainly driven by the hc (86%) but can you give the % of each of the 3 domain of the hc contribution please?

P5-6

In MHC-I there is a conserved Lys146 that contact the Cter of the peptide binding to the F pocket, is there interaction between this MHC residue and the editing loop? A proposition would be to report the contacts between the molecules as a supp table.

P7

Even if the sequence alignment of diverse MHC and MHC-like molecule suggests that other molecules could bind tapasin, why would be the role of tapasin for non-peptide binding molecules like MR1 or CD1a?

Given the structure of the complex (Fig 5), and the focus on the Cterminal part of the cleft, this suggests that the residue at the F pocket might binds first, and then the rest of the peptide fit. How does this model works for peptides with extended Cterminal tail?

Structure of an MHC I–tapasin–ERp57 editing complex defines chaperone promiscuity
 NCOMMS-22-04150-T by Müller et al.

Point-by-point response

We wish, at the outset, to thank the reviewers for their very strong and generous support of our work, and for their many helpful suggestions, which have improved the manuscript.

Reviewer #1:

The structure of an MHC class I complex with tapasin is important for the field, even though there are few surprises from this structure, given the recent structures of the related complexes of MHC class I with TAPBPR (Jiang et al 2017 and Thomas and Tampe, 2017). Nonetheless, it is great to see this structure.

Reply: We wish to thank the reviewer for the very positive overall evaluation of our work and for sharing his/her perspective on the manuscript.

Some comments & suggestions:

- 1) *page 4 and page 7: Escape pathway-explain what this is.*

Reply and Action Taken: In the introduction of the revised manuscript, we have explained the escape pathway in more detail. Now, the text reads: “The interaction between tapasin and ERp57 relies on a mixed disulfide between Cys95 of tapasin and Cys33 of ERp57 (Cys57 in immature ERp57)⁴. In the oxidizing environment of the ER, Cys33 of ERp57 toggles between an intermolecular, mixed disulfide with Cys95 of tapasin and an intramolecular disulfide with Cys36 (Cys60 in immature ERp57)¹⁰. This toggling, known as the escape pathway, has been described for protein-disulfide isomerase-assisted protein folding and prevents the enzyme from becoming trapped in covalent complexes with its substrates¹¹.”

- 2) *Page 5: "Furthermore, the side chain of Leu18 of the editor loop, which was shown to be essential for MHC I surface expression 28,29,30," It appears that the leucine is not important for some MHC-I allotypes including H2-Db (Hafstrand et al).*

Reply and Action Taken: We apologize for this generalized statement. We have rephrased the sentence as follows: “Furthermore, the side chain of Leu18 of the editor loop, which was shown to be relevant for MHC I surface expression, compensates in the peptide-free MHC I F pocket for the lack of a C-terminal hydrophobic side chain of the antigenic peptide and presumably assists in groove widening.”

- 3) *Since a large part of the discussion relates to the "editing loop", it would be helpful in Figure 3 to have data that unambiguously demonstrates the functional relevance of this loop to H2-Db assembly.*

Reply and Action Taken: For the tapasin homologue TAPBPR the functional relevance of the editing loop has been demonstrated for H2-D^b (Sagert *et al.* 2020 *eLife*). A mutant of TAPBPR with three glycine residues replacing the editing loop showed a decreased complex stabilization with H2-D^b compared to wildtype TAPBPR. Also, the mutant lacking the editing loop did not show any activity in an *in vitro* peptide exchange assay even at high concentrations (Sagert *et al.* 2020 *eLife*).

To demonstrate the functional relevance of the tapasin editing loop, we replaced the loop for three glycine residues (Δ loop) or with the editing loop of TAPBPR (loop^{TAPBPR}) (Fig. R1a). As for the other tapasin mutants shown in the manuscript, we utilized a construct that links expression of an enhanced green fluorescent protein (GFP) reporter to the expression of the mutant protein via an internal ribosomal entry site. GFP-positive cells were analyzed for MHC I surface expression by flow cytometry. GFP-positive cells were probed by immunoblotting for expression of mutant tapasin. The Δ loop construct showed a reduction of surface MHC I to 60±4% compared to wildtype tapasin

(Fig. R1b). The substitution with the editing loop of TAPBPR resulted in a slightly higher MHC I surface expression of $71 \pm 2\%$ but could not compensate for the editing loop of tapasin (Fig. R1b). However, the expression levels of both loop mutants were rather low compared to wildtype (Fig. R1c). Since we could not rule out that reduced MHC I surface presentation resulted from loop mutants or was due to lower expression levels of the tapasin mutants, when compared to wildtype tapasin, we did not include these data in the manuscript.

Figure R1: **Effects of editing loop mutants of tapasin on MHC I surface expression.** **a**, Sequence alignment of editing loop region of tapasin, TAPBPR and tapasin mutants. **b**, MHC I surface expression of tapasin-deficient HAP1 cells, expressing wildtype (WT) and loop mutants of tapasin. Flow cytometry was conducted using an APC-labeled HLA-A/B, C-specific antibody (W6/32). The upper dashed line represents the MHC I surface level of cells, expressing wildtype tapasin. The lower dashed line represents MHC I surface expression of cells transfected with a vector devoid of gene of interest (mock). The mean fluorescent intensities were normalized to wildtype tapasin (\pm SD; $n=3$ biologically independent samples). The gating strategy is provided in Supplementary Fig. 8. **c**, Immunoblots of whole cell extracts of GFP-positive cells of wildtype and loop mutants of tapasin (α -tapasin, α -GAPDH). -, mock transfection.

- 4) *Since the C-domain of tapasin forms a significant interface with β_2m , it would be useful to have some mutagenesis studies focused on that region.*

Reply and Action Taken: We identified three hydrogen bonds between the C-terminal domain of tapasin with β_2m . Two of the contacts are formed by backbone interactions of Leu329 of tapasin with Ile92 of β_2m and one side chain interaction of Glu307 of tapasin with Ser88 of β_2m . We generated the mutant Glu307Ala and included the data in Fig. 4e and Supplementary Fig. 3e. Text on page 6 was adjusted accordingly: "Disruption of the hydrogen bond between Glu307 of tapasin and Ser88 of β_2m reduced the MHC I surface level to $46 \pm 1\%$ (Fig. 4e,f); albeit the expression level of mutant tapasin was also lowered compared to WT (Supplementary Fig. 3e)".

- 5) *Page 6 "Because MHC I F-pocket residues Thr143, Lys146, and Trp147, which coordinate the C terminus of bound peptides, are not blocked by residues of the editor loop (Supplementary Fig. 3a), these data suggest that the editor loop does not act as a peptide surrogate as discussed for TAPBPR,27,31,32." The authors are referring to many prior publications from the Tampe lab which have highlighted a competitor/peptide surrogate role for the scoop loop of TAPBPR in TAPBPR-mediated editing, which has been controversial in the literature. They should clarify whether they are suggesting the mechanism of function of the editing/scoop loop is different in TAPBPR vs tapasin, and if so why. Alternatively, are they suggesting that their prior findings should be more cautiously interpreted? The parallel discussions on page 7 should also be clarified.*

Reply and Action Taken: We revised the discussion for clarity. We propose different peptide editing mechanisms for tapasin and its homologue TAPBPR. The editing loop of tapasin is shorter and cannot adopt a conformation in which it dives into the binding-pocket of MHC I, allowing it to act as a surrogate for a peptides C-terminus as seen for TAPBPR (Thomas & Tampé 2017 *Science*; Sagert *et al.* 2020 *eLife*). While tapasin as part of the PLC acts in a peptide-rich environment of the ER, TAPBPR has been found to edit MHC I in a peptide-depleted environment as the *Golgi* compartment. Therefore, the longer loop of TAPBPR may help stabilizing peptide-deficient MHC I molecules for longer periods of time. We have changed the text accordingly: "This is because MHC I F-pocket residues Asn80, Tyr84, and Lys146, which

coordinate the terminal carboxyl group of bound peptides, are engaged in interactions with tapasin and its editing loop (Fig. 3a,b and Supplementary Fig. 4a). Our data suggest that tapasin acts in a dual manner, opening the peptide binding groove and stabilizing the empty F-pocket of the MHC I hc.”

- 6) *Overall, the presentation of Figures 3 and 4 could be improved by highlighting the specific region of the full complex that is in focus for each of those figures. Figure 5 (left panel) is also suggesting a peptide competitor model for the editing loop, which could be modified.*

Reply and Action Taken: We wish to thank the reviewer for her/his suggestion. We modified the figures 3 and 4 accordingly, which now highlight the specific regions in relation to the full MHC I chaperone complex. Additionally, we have modified Figure 5 and modified the figure legend, which now reads: “Schematic of molecular motions leading to stabilized peptide-receptive clients upon association with tapasin (left, side view; right, top view). Conformational changes are indicated by black arrows. The dotted line indicates a polar interaction between tapasin (Glu⁷²) and chaperoned MHC I (Tyr⁸⁴ of the heavy chain, hc). ERp57 is not shown in the right panel for reasons of clarity.”

Reviewer #2:

The chaperone tapasin in conjunction with the oxidoreductase ERp57 is crucial for MHC I assembly and for shaping the epitope repertoire for high immunogenicity. The manuscript by I. K. Müller et al. resolve a crystal structure of a tapasin–ERp57 heterodimer in complex with a peptide-receptive MHC I heterodimer at 2.7 Å resolution, representing a transient ER quality control complex. They show the editing loop of tapasin rigidifies upon client recognition and contributes to a widened MHC I F-pocket, and suggest elements indispensable for peptide proofreading. Overall, the structural and functional work gives new insight into the mechanistic basis for the selector function of tapasin and reveal the level of plasticity of peptide-receptive MHC I determines chaperone-assisted acquisition of high-affinity peptides. The following are comments and questions the authors need to address.

Reply: We would like to thank the reviewer for these encouraging comments highlighting the new mechanistic insight into the function of tapasin and MHC I chaperone complexes.

- 1) *It is not clear how the authors assembled the MHC I–tapasin–ERp57 editing complex by a photo-triggered approach. Please add more detailed introduction on the photo-triggered approach, and how did they induce the formation of the transient complex? Did they resolve the bound peptide in the crystal structure? If not, please explain why the peptide was not captured.*

Reply and Action Taken: In analogy to our structural analysis of the TAPBPR-MHC I chaperone complex (Thomas and Tampé, 2017, *Science*) we captured peptide-receptive MHC I-tapasin-ERp57 complexes by photoablation of photo-sensitive MHC I-bound peptides. We utilized such photo-triggered approach to generate a peptide-receptive MHC I molecule. The photocleavage of the backbone of the peptide leads to the formation of two peptide fragments each with a drastically decreased affinity to the MHC I peptide-binding groove.

To show that photo-cleavage leads to peptide-receptive MHC I, we have performed size exclusion chromatography-coupled mass spectrometry (SEC-MS) analyses of UV-irradiated MHC I molecules and included these data as new Supplementary Fig. 2. The experiments demonstrate that less than 4.0% of MHC I carry the intact peptide (RGPGRFJ*TI) after UV irradiation and only 1.8% the larger photo-cleaved fragment (RGPGRF). The partial association of uncleaved peptide with MHC I agrees with the residual electron density in the peptide-binding groove of MHC I that may represent intact or fragmented peptides. We would like to stress that the occupancy of such potential peptide fragments is very low (< 10%) and the residual difference density did not allow unambiguous modeling of an uncleaved peptide or a peptide fragment in the MHC I binding pocket. Notably, the crystallization condition contained high concentrations (100 mM) of the dipeptide Gly-Gly, which may account for the diffuse residual density in the peptide binding groove.

We have updated the text on page 5: “Size exclusion chromatography-coupled mass spectrometry (SEC-MS) revealed an almost complete photoinduced cleavage (>96%) of the peptide bound to MHC I (Supplementary Fig. 2).”

- 2) *The authors describe on page 4 that “We utilized the C60A substitution in ERp57 to prevent the disulfide cleavage via the escape pathway”. Please add introduction on the escape pathway, and also explain why used the C60A substitution in ERp57 to prevent the disulfide cleavage via the escape pathway.*

Reply and Action Taken: To clarify the reviewers comment we have included a short description of the escape pathway in the introduction of the revised manuscript that reads: “The interaction between tapasin and ERp57 relies on a mixed disulfide between Cys95 of tapasin and Cys33 of ERp57 (Cys57 in immature ERp57)⁴. In the oxidizing environment of the ER, Cys33 of ERp57 toggles between an intermolecular, mixed disulfide with Cys95 of tapasin and an intramolecular disulfide with Cys36 (Cys60 in immature ERp57)¹⁰. This toggling, known as the escape pathway, has been described for protein-disulfide isomerase-assisted protein folding and prevents the enzyme from becoming trapped in covalent complexes with its substrates¹¹.”

Furthermore, we have added the following sentence to the first paragraph of the results section: “Furthermore, we substituted Cys33 of ERp57 with Ala (C36A), formerly described as C60A in immature ERp57 (ref.⁴), to retain the mixed disulfide between tapasin and ERp57 (ref.¹⁰).

- 3) *Regarding the crystal structure, the clashscore appears relatively high, and the structure need to be further refined and improved.*

Reply and Action Taken: We have utilized the PDB-REDO server with additional subsequent manual refinement runs that improved model statistics. We substantially improved the clashscore to 5.46 with an overall MolProbity score of 1.64 retaining good stereochemistry values. We have updated the methods section and Supplementary Table 1, accordingly, and updated the PDB deposition.

- 4) *On page 4, the authors describe that “Disulfide reduction does not result from UV illumination performed for photocleavage of the peptide but is rather facilitated by X-ray irradiation (Supplementary Fig. 1e,f). Therefore, we speculate that the escape pathway is utilized to switch between a non-editing and an editing conformation of the tapasin–ERp57 heterodimer”. Please explain why the disulfide reduction is facilitated by X-ray irradiation, and how did the authors deduce the speculation “that the escape pathway is utilized to switch between a non-editing and an editing conformation of the tapasin–ERp57 heterodimer”.*

Reply: We have shown that the disulfide bond was not reduced by UV illumination (Supplementary Fig. 1). We speculated that the reduction resulted either from X-ray irradiation or from the editing process itself that may pose a molecular tension on the ERp57-tapasin interaction and in particular on the disulfide bond the covalently links ERp57 to tapasin. Radiation-induced damage to protein crystals, specifically to disulfide bonds, during X-ray diffraction data collection is well described phenomenon. Not all disulfide bridges are equally susceptible to radiation damage (Bhattacharyya et al. 2020 IUCrJ). Disulfide bonds, which are in close contact with a carbonyl O atom aligned with the S–S bond for a nucleophilic attack, are more susceptible to the reduction. In order not to overinterpret our data we have merged the first two subsections of the results part which now concludes on page 4: “Since disulfide bond reduction does not result from UV illumination performed for photocleavage of the peptide (Supplementary Fig. 1), we speculate that reduction is either facilitated by X-ray irradiation²⁷ or may result from the MHC I editing process itself.”

- 5) *The authors mention on page 5 that “..... a loop comprising the residues Glu11-Lys20 (referred to as editing loop) sits on top of the F-pocket”, however it was later on referred to as “editor loop”. Please clarify the loop should be referred to as “editing loop” or “editor loop”.*

Reply and Action Taken: We wish to thank the reviewer for spotting this inconsistency. We have now used the term “editing loop” throughout the manuscript.

- 6) *The authors are suggested to give a more complete explanation on Fig. 5 in the text.*

Reply and Action Taken: Excellent suggestion. We have now provided an extended explanation on Fig. 5 in the discussion: “In essence, the crystal structure of the MHC I–tapasin–ERp57 editing complex reveals that both tapasin-dependent and -independent allomorphs have the structural prerequisites to interact with the chaperone, suggesting that their degree of plasticity and hence their ability to accommodate high-affinity peptide cargo determines the association, which is stabilized by key structural elements of tapasin (Fig. 5). It needs further structural and functional characterization to understand how *e.g.* C-terminal extended peptides that protrude from the MHC I F-pocket are edited by tapasin or TAPBPR. As tapasin acts primarily as part of the PLC multichaperone network, we speculate that association of ERp57 with tapasin is not only necessary for coordinating the glycan-chaperone calreticulin, but that the molecular dynamics of this interaction may help unlock tapasin for client engagement and peptide proofreading catalysis.”

Reviewer #3:

Muller and colleagues present here the structure of MHC-I in complex with tapasin and ERp57 that show the fine mechanism of peptide loading, a critical step for the immune sensing of microbial infection. The manuscript is well written, clear, and the data excellent.

The complex is only transient in the cell, so being able to crystallize the complex is a tour de force and required the use of photo-triggered technique. The structure reveals 2 loops that are driving the key interaction, one scooping the antigen binding cleft to stabilize and facilitate peptide exchange, and one that ensure the stability like a “firm hand” underneath the MHC that will support the fold of the protein known to be unstable if empty. Previously, the structure of HLA-II in complex with the HLA-DM molecule showed how peptide can go in and out of the cleft in HLA-II without the all “empty” MHC-II falling apart, we finally understand how different molecules (Tapasin and ERp57) work in concert to stabilize and swap peptide in the cleft.

Thanks for the movie it’s very helpful to visualize the interaction between the molecules.

Reply: We highly appreciate that reviewer #3 shares our opinion about the excellent data in the current manuscript. We would like to thank the reviewer for the helpful and supportive comments.

Comments & questions:

- 1) *Page 2: According to the <https://www.ebi.ac.uk/ipd/imqt/hla/about/statistics/> web site there are 23417 HLA-A/B/C alleles identified so far, probably worth updating the number reported in the introduction.*

Reply and Action Taken: We wish to thank the reviewer. The updated numbers for HLA class I alleles (~24,000) are included in the revised manuscript.

- 2) *Page 4: What other MHC, beside H2Db, were screened and what was the difference with H2Db? Would the authors have an explanation for why H2Db was better for structural determination?*

Reply: Besides the mouse MHC I H2-D^b, we screened various human allomorphs that were described to be tapasin-dependent (Rizvi *et al.* 2014 *J Immunol*, Bashirova *et al.* 2020 *PNAS*) and/or were found to be sampled by the PLC (Blees *et al.* 2017 *Nature*). Crystal structure analysis comparing a human $\beta_2m/H2-D^b$ to a murine $\beta_2m/H2-D^b$ complex shows increased polarity and number of hydrogen bonds to the $\alpha1\alpha2$ domains of H2-D^b with human β_2m (Achour *et al.* 2006 *J Mol Biol*). This stabilized interface allows for a prolonged peptide-receptive conformation of H2-D^b and presumably helps tapasin to engage empty MHC-I in a kinetically more favorable fashion.

- 3) *Page 5: The total surface area between the tapasin and MHC-I is mainly driven by the hc (86%) but can you give the % of each of the 3 domain of the hc contribution please?*

Reply and Action Taken: The buried surface area (BSA) between tapasin and MHC I covers 4,381 Å² (hc: 74%; β_{2m}: 26%). The α1 and α2 domains form 2,625 Å² (52% of BSA of MHC I), while the α3 domain contributes with 624 Å² (19%). We have updated the revised manuscript on pages 4 and 5 as: “The interface between the N-terminal domain of tapasin and MHC I covers a total surface area of 4,381 Å² (hc: 74%; β_{2m}: 26%) (Supplementary Fig. 3) of which the N-terminal domain of tapasin contributes with 2,657 Å².” and “Furthermore, the C-terminal Ig-like domain of tapasin swings towards the α3 domain of the MHC I hc and β_{2m} (Fig. 2d), allowing for an additional interaction interface with a buried surface area of 1,724 Å² (hc: 55%; β_{2m}: 45%; Supplementary Fig. 3a,b).”

- 4) *Page 5-6: In MHC-I there is a conserved Lys146 that contact the Cter of the peptide binding to the F pocket, is there interaction between this MHC residue and the editing loop? A proposition would be to report the contacts between the molecules as a supp table.*

Reply and Action Taken: We wish to thank the reviewer for this comment. Yes, there is a salt bridge, which is formed by Lys146 of MHC I and Asp12 of the editing loop of tapasin. We included this interaction in figure 3 and updated the text on page 5 as: “The side chain of Asp12 of tapasin is in salt-bridge distance to the ε-amino group of Lys146 of the MHC I hc that presumably contributes to the widening of the peptide-binding groove and the rigidification of the editing loop of tapasin (Fig. 3d).”

- 5) *Page 7: Even if the sequence alignment of diverse MHC and MHC-like molecule suggests that other molecules could bind tapasin, why would be the role of tapasin for non-peptide binding molecules like MR1 or CD1a?*

Reply: We agree with reviewer #3 that it is unlikely that tapasin edits MR1 and CD1a because their ligands are non-peptides. Since it is not known where and how MR1 and CD1a are loaded with their substrates, we removed MR1 and CD1a from the main text but included these non-classical MHC-related molecules for the sake of completeness.

- 6) *Given the structure of the complex (Fig. 5), and the focus on the C-terminal part of the cleft, this suggests that the residue at the F pocket might binds first, and then the rest of the peptide fit. How does this model work for peptides with extended C-terminal tail?*

Reply and Action Taken: The reviewer addresses an interesting point here. Superposition of HLA-A*02:01 carrying a C-terminally extended peptide (PBD ID: 2CLR) and our MHC I-tapasin-ERp57 structure allows a chaperoned MHC I complex when bound to a C-terminally extended peptide, indicating that also a C-terminally extended peptide could bind first at the F-pocket and then the rest of the peptide fit. We extended the discussion with the following sentence: “It needs further structural and functional characterization to understand how e.g. C-terminally extended peptides that protrude from the MHC I F-pocket are edited by tapasin or TAPBPR.”

REVIEWERS' COMMENTS

Reviewer #1 (Remarks to the Author):

I appreciate many edits in line with previous suggestions, but I still find the presentation of the editing loop data quite confusing, and there is no clear-cut discussion related to differences between the editing loops of tapasin and TAPBPR.

For example:

"Furthermore, the side chain of Leu18 of the editing loop, which was shown to be relevant for MHC I surface expression³⁰⁻³², compensates in the peptide-free MHC I F-pocket for the lack of a C-terminal hydrophobic side chain of the antigenic peptide and presumably assists in groove widening. In line with this notion, substitution of Leu18 with a bulky hydrophobic side chain (Trp) did not negatively impact MHC I surface presentation in flow cytometry-based cellular assays when compared to the wildtype (WT) (Fig. 3e and Supplementary Fig. 3c). Conversely, substitution of Leu18 with glycine was shown to nearly abolish peptide exchange activity of tapasin³²."

A) These statements are imprecise about whether the residue (L18) acts or does not act as a peptide surrogate by inserting inside the groove. Figure 3C and the models of Figure 5 do little to clarify this point and could be misleading.

B) Additionally, the last two sentences compare different sets of experiments done with the Trp vs. Gly mutants in two different studies, while no data are provided about whether the Gly mutant in fact will function differently than the Trp mutant in the analyses of Figure 3e. Yet, some conclusions are attempted based on the two studies and two sets of mutants.

C) Furthermore, the cited references (30-32) include both TAPBPR and tapasin, but do not differentiate the potential functions and sizes of the two editing loops or provide related clarity.

Again, these points have been controversial in the literature and providing better clarity on potential differences between their tapasin and TAPBPR structures is important for this new study and for readers.

Reviewer #2 (Remarks to the Author):

The authors have addressed my comments/questions in the satisfactory manner. I would recommend publication of the manuscript.

Reviewer #3 (Remarks to the Author):

The authors have answered all my questions and accordingly revised the manuscript. I have no further comments.

Structure of an MHC I–tapasin–ERp57 editing complex defines chaperone promiscuity NCOMMS-22-04150-T by Müller et al.

Point-by-point response

Reviewer #1:

I appreciate many edits in line with previous suggestions, but I still find the presentation of the editing loop data quite confusing, and there is no clear-cut discussion related to differences between the editing loops of tapasin and TAPBPR.

For example: "Furthermore, the side chain of Leu18 of the editing loop, which was shown to be relevant for MHC I surface expression³⁰⁻³², compensates in the peptide-free MHC I F-pocket for the lack of a C-terminal hydrophobic side chain of the antigenic peptide and presumably assists in groove widening. In line with this notion, substitution of Leu18 with a bulky hydrophobic side chain (Trp) did not negatively impact MHC I surface presentation in flow cytometry-based cellular assays when compared to the wildtype (WT) (Fig. 3e and Supplementary Fig. 3c). Conversely, substitution of Leu18 with glycine was shown to nearly abolish peptide exchange activity of tapasin³²."

A) These statements are imprecise about whether the residue (L18) acts or does not act as a peptide surrogate by inserting inside the groove. Figure 3C and the models of Figure 5 do little to clarify this point and could be misleading.

To A) Our structural data show that Leu18 sits near the location of a hydrophobic side chain at the C terminus of a bound peptide. It thereby does not act as a peptide surrogate per se. In the absence of bound peptide, the loop rather assists in groove widening. We assume that the plasticity of the loop structure allows remodeling when the F-pocket is occupied by a peptide. This dynamic character may help stabilizing bound peptides, as described by the molecular trap hypothesis. In the current manuscript, we have therefore postulated that the loop serves peptide exchange in a dual fashion. We have added the following sentence to clarify the reviewer's concern and to better specify our statements for the reader: "Even though sitting directly above the F-pocket, Leu18 does not mimic the C terminus of cargo peptides. Thus, Leu18 is unlikely to act as a peptide surrogate per se³⁰."

B) Additionally, the last two sentences compare different sets of experiments done with the Trp vs. Gly mutants in two different studies, while no data are provided about whether the Gly mutant in fact will function differently than the Trp mutant in the analyses of Figure 3e. Yet, some conclusions are attempted based on the two studies and two sets of mutants.

To B) We have stressed the difference of the two studies by slightly changing the text, which now reads: "In line with this groove-widening function, substitution of Leu18 with a bulky hydrophobic side chain (tryptophan) did not negatively impact MHC I surface presentation in our flow cytometry-based cellular assays when compared to the wildtype (WT) (Fig. 3e and Supplementary Fig. 3c). Conversely, substitution of Leu18 with glycine has been shown to nearly abolish peptide exchange activity of tapasin³¹."

C) Furthermore, the cited references (30-32) include both TAPBPR and tapasin, but do not differentiate the potential functions and sizes of the two editing loops or provide related clarity

To C) We have removed ref.30 that related to TAPBPR from the main text.